# iShape: A First Step Towards Irregular Shape Instance Segmentation

**Lei Yang**
yanglei@megvii.com

**Ziwei Yan**
yanzw@buaa.edu.cn

**Wei Sun**
sunwei@megvii.com

**Yisheng He**
yhebk@connect.ust.hk

**Zhenhang Huang**
zhhuang@buct.edu.cn

**Haibin Huang**
jackiehuanghaibin@gmail.com

**Haoqiang Fan**
fhq@megvii.com

## Abstract

In this paper, we introduce a brand new dataset to promote the study of instance segmentation for objects with irregular shapes. Our key observation is that though irregularly shaped objects widely exist in daily life and industrial scenarios, they received little attention in the instance segmentation field due to the lack of corresponding datasets. To fill this gap, we propose iShape, an irregular shape dataset for instance segmentation. Unlike most existing instance segmentation datasets of regular objects, iShape has many characteristics that challenge existing instance segmentation algorithms, such as large overlaps between bounding boxes of instances, extreme aspect ratios, and large numbers of connected components per instance. We benchmark popular instance segmentation methods on iShape and find their performance drop dramatically. Hence, we propose an affinity-based instance segmentation algorithm, called ASIS, as a stronger baseline. ASIS explicitly combines perception and reasoning to solve **A**rbitrary **S**hape **I**nstance **S**egmentation including irregular objects. Experimental results show that ASIS outperforms the state-of-the-art on iShape. Dataset and code are available at http://ishape.github.io

## 1 Introduction

Instance segmentation aims to predict the semantic and instance labels of each image pixel. Compared to object detection [1, 2, 3, 4, 5, 6, 7, 8] and semantic segmentation [9, 10, 11], instance segmentation provides more fine-grained information but is more challenging and attracts more and more research interests of the community. Many methods [12, 13, 14, 15] and datasets [16, 17, 18] continue to emerge in this field. However, most of them focus on regularly shaped objects and only a few [19, 18] study irregular ones, which are thin, curved, or having complex boundary and can not be well-represented by regularly rectangular boxes. We think the insufficient explo-

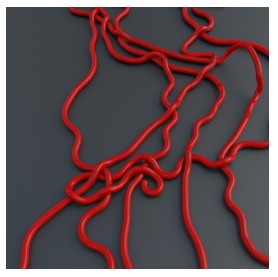
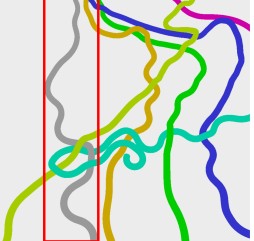

(a) iShape-Wire      (b) Ground Truth

Figure 1: A typical scene of objects with irregular shape and similar appearance. It has many characteristics that challenge instance segmentation algorithms, including the large overlaps between bounding boxes of objects, extreme aspect ratios (bounding box of the grey mask), and large numbers of connected components in one instance (green and blue masks).

Submitted to the 35th Conference on Neural Information Processing Systems (NeurIPS 2021) Track on Datasets and Benchmarks. Do not distribute.

ration of this direction is caused by the lack
of corresponding datasets.

In this work, we present iShape, a new dataset designed for **i**rregular **Shape** instance segmentation. Our dataset consists of six sub-datasets, namely iShape-Antenna, iShape-Branch, iShape-Fence, iShape-Log, iShape-Hanger, and iShape-Wire. As shown in Figure 2, each sub-dataset represents scenes of a typical irregular shape, for example, strip shape, hollow shape, and mesh shape. iShape has many characteristics that reflect the difficulty of instance segmentation for irregularly shaped objects. The most prominent one is the large overlaps between bounding boxes of objects, which is hard for proposal-based methods[12, 14] due to feature ambiguity and non-maximum suppression (NMS [20]). Meanwhile, overlapped objects that share the same center point challenge center-based methods[21, 22, 23]. Another characteristic of iShape is a large number of objects with similar appearances, which makes embedding-based methods[24, 25] hard to learn discriminative embedding. Besides, each sub-dataset has some unique characteristics. For example, iShape-Fence has about 53 connected components per instance, and iShape-Log has a large object scale variation due to various camera locations and perspective transformations. We hope that iShape can serve as a complement of existing datasets to promote the study of instance segmentation for irregular shape as well as arbitrary shape objects.

We also benchmark existing instance segmentation algorithms on iShape and find their performance degrades significantly. To this end, we introduce a stronger baseline considering irregular shape in this paper, which explicitly combines perception and reasoning. Our key insight is to simulate how a person identifies an irregular object. Taking the wire shown in Figure 1 for example, one natural way is to start from a local point and gradually expand by following the wire contour and figure out the entire object. The behavior of such "following the contour" procedure is a process of **continuous iterative reasoning based on local clues**, which is similar to the recent affinity-based approaches [26, 27]. Under such observation, we propose a novel affinity-based instance segmentation baseline, called ASIS, which includes principles of generating effective and efficient affinity kernel based on dataset property to solve **A**rbitrary **S**hape **I**nstance **S**egmentation. Experimental results show that the proposed baseline outperforms existing state-of-the-art methods by a large margin on iShape.

Our contribution is summarized as follows:

- We propose a brand new dataset, named iShape, which focuses on irregular shape instance segmentation and has many characteristics that challenge existing methods. In particular, we analyzed the advantages of iShape over other instance segmentation datasets.

- We benchmark popular instance segmentation algorithms on iShape to reveal the drawbacks of existing algorithms on irregularly shaped objects.

- Inspired by human's behavior on instance segmentation, we propose ASIS as a stronger baseline on iShape, which explicitly combines perception and reasoning to solve Arbitrary Shape Instance Segmentation.

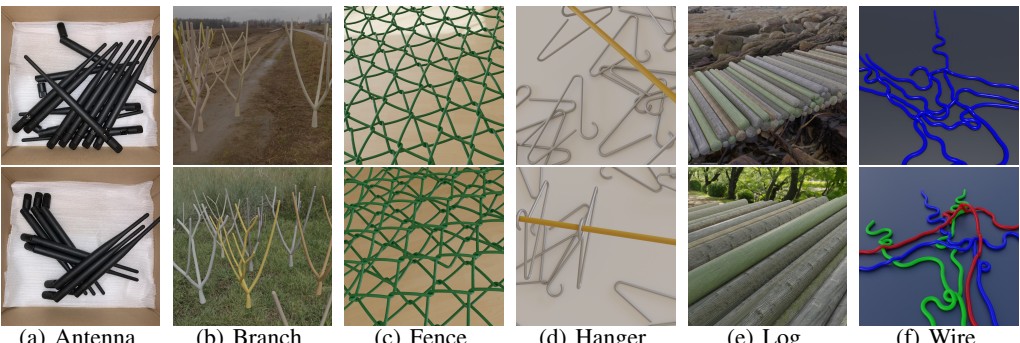

| (a) Antenna | (b) Branch | (c) Fence | (d) Hanger | (e) Log | (f) Wire |

Figure 2: The six sub-datasets in iShape.

## 2   Related Work

### 2.1   Existing Datasets

There are several benchmark datasets collected to promote the exploration of instance segmentation. The COCO [16] and the Cityscapes [17] are the most popular ones among them. However, the shapes of target objects in these datasets are too regular. The connected components per instance (CCPI) and average MaxIoU are low in the datasets and state-of-the-art algorithms selected from them can not generalize to more challenging scenarios. Instead, in the scenario of human detection and segmentation, the OC human [19] and the Crowd Human [28] introduce datasets with larger MaxIoU. Nevertheless, the OC human dataset only provides a small number of images for testing, and the number of instances per image is too small to challenge instance segmentation algorithms. While the crowd human dataset only provides annotations of object bounding boxes, limiting their application to the instance segmentation field. In the area of photogrammetry, the iSAID [18] dataset is proposed to lead algorithms to tackle objects with multi scales. However, shapes of objects in this dataset are common, most of which are rectangular, and the lack of instance overlapping reduces its challenge to instance segmentation algorithms as well. Under the observation that these existing regular datasets are not enough to challenge algorithms for more general scenarios, we propose iShape, which contains irregularly shaped objects with large overlaps between bounding boxes of objects, extreme aspect ratios, and large numbers of CCPI to promote the capabilities of instance segmentation algorithms.

### 2.2   Instance Segmentation Algorithms

Existing instance segmentation algorithms can be divided into two classes, proposal-based and proposal-free.

**Proposal-based approaches** One line of these approaches [12, 14, 29] solve instance segmentation within a two-stage manner, by first propose regions of interests (RoIs) and then regress the semantic labels of pixels within them. The drawback of these approaches comes from the loss of objects by NMS due to large IoU. Instead, works like [15] tackle the problem within a single-stage manner. For example, PolarMask [15] models the contours based on the polar coordinate system and then obtain instance segmentation by center classification and dense distance regression. But the convex hull setting limits its accuracy.

**Proposal-free approaches** To shake off the rely on proposals and avoid the drawback caused by them, many bottom-up approaches like [22, 23, 24, 25] are introduced. These works are in various frameworks. The recent affinity-based methods obtain instance segmentation via affinity derivation [26] and graph partition[30]. This formulation is more similar to the perception and reasoning procedure of we human beings and can handle more challenging scenarios. GMIS [26] utilizes both region proposals and pixel affinities to segment images and SSAP [27] outputs the affinity pyramid and then performs cascaded graph partition. However, The affinity kernels of GMIS and SSAP are sparse in angle and distance, leading to missing components of some instances due to loss of affinity connection. To this end, we propose ASIS which includes principles of generating effective and efficient affinity kernel based on dataset property to solve Arbitrary Shape Instance Segmentation and achieve great improvement on iShape.

## 3   iShape Dataset

### 3.1   Dataset Creation

iShape consists of six sub-datasets. One of them, iShape-Antenna, is collected from real scenes, which are used for antenna counting and grasping in automatic production lines. The other five sub-datasets are synthetic datasets that try to simulate five typical irregular shape instance segmentation scenes.

**iShape-Antenna Creation.** For the creation of iShape-Antenna, we first prepare a carton with a white cushion at the bottom, then randomly and elaborately place antennas in it to generate various scenes. Above the box, there is a camera with a light that points to the inside of the box to capture the scene images. We collect 370 pictures and annotate 3,036 instance masks then split them equally for training and testing. The labeling is done by our supplier. We have checked all the annotations

 Although iShape-Antenna only contains 370 images, the number of instances reaches 3,036 which is more than most categories in Cityscapes [17] and PASCAL VOC [31].

**Synthetic Sub-datasets Creation** There are lots of typical irregular shape instance segmentation scenes. Consequently, it is impractical to collect a natural dataset for each typical scene. Since it is traditional to study computer vision problems using synthetic data [32, 33], we synthesize five sub-datasets of iShape which include iShape-Branch, iShape-Fence, iShape-Log, iShape-Hanger, and iShape-Wire, by using CG software Blender. In particular, We build corresponding 3D models and placement they appropriate in Blender with optional random background and lighting environment, optional physic engine, and random camera position. The creation configs of synthesis sub-datasets are listed in the appendix. After setting up the scene, we use a ray tracing render engine to render the RGB image. Besides, We build and open source a blender module, bpycv [34], to generate instance annotation. We generate 2500 images for each sub-dataset, 2000 for training, 500 for testing.

## 3.2 Dataset Characteristics

In this sub-section, we analyze the characteristics of iShape and compare it with other instance segmentation datasets. Since each sub-dataset represents irregularly shaped objects in different scenes, we present the statistical results of each sub-dataset separately.

**Dataset basic information.** As summarized in Table 1, iShape contains 12,870 images with 175,840 instances. All images are $1024 \times 1024$ pixels and annotated with pixel-level ground truth instance masks. Since iShape focus on evaluating the performance of algorithms on the irregular shape, each scene consists of multiple instances of one class, which is also common cases in industrial scenarios.

**Instance count per image.** A larger instance count is more challenging. Despite iSAID getting the highest instance count per image, it is unfair for extremely high-resolution images and normal-resolution images to be compared on the indicator. Among iShape, the instance count per image of iShape-Log reaches 28.86 that significantly higher than other normal-resolution datasets.

**The large overlap between objects.** We introduce a new indicator, Overlap of Sum (OoS), which aims to measure the degree of occlusion and crowding in a scene, defined as follows:

$$Overlap\ of\ Sum = \begin{cases} 1 - \frac{|\bigcup_{i=1}^{n} C_i|}{\sum_{i=1}^{n} |C_i|}, & n > 0 \\ 0, & n = 0 \end{cases} \quad (1)$$

where $C$ means bounding boxes(bbox) or convex hulls(convex) of all instances in the image, $n$ means number of instances, $\bigcup$ means union operation, and $|C_i|$ means to get the area of $C_i$. The statistics of average OoS for bounding box and convex hull are presented in Table. 1. For bounding box OoS, All iShape sub-datasets are higher than other datasets, which reflects the large overlap characteristic of iShape. Thanks to the large-area hollow structure, iShape-Fence gets the highest average convex hull OoS 0.63. Moreover, The Average MaxIoU [19] of all images also reflects the large overlap characteristic of iShape.

Table 1: Comparison of statistics with different datasets.

| Dataset | Images | Ins. | Ins./image | OoS | | AvgMIoU | Aspect ratio | CCPI |
|---------|--------|------|-----------|------|--------|---------|------|------|
| | | | | bbox | convex | | | |
| Cityscapes | 2,975 | 52,139 | 17.52 | 0.14 | 0.07 | 0.394 | 2.29 | 1.34 |
| COCO | **123,287** | **895,795** | 7.26 | 0.15 | 0.09 | 0.210 | 2.59 | 1.41 |
| CrowdHuman | 15,000 | 339,565 | 22.64 | - | - | - | - | - |
| OC Human | 4,731 | 8,110 | 1.71 | 0.25 | 0.20 | 0.424 | 2.28 | 3.11 |
| iSAID | 2,806 | 655,451 | **233.58** | - | - | - | 2.40 | - |
| Antenna | 370 | 3,036 | 8.20 | 0.62 | 0.23 | 0.655 | 9.86 | 2.45 |
| Branch | 2,500 | 26,046 | 10.14 | 0.62 | 0.52 | 0.750 | 2.47 | 10.88 |
| Fence | 2,500 | 7,870 | 3.15 | 0.65 | **0.63** | **0.983** | 1.05 | **53.65** |
| Hanger | 2,500 | 49,275 | 19,71 | 0.53 | 0.34 | 0.685 | 3.28 | 4.94 |
| Log | 2,500 | 72,144 | 28.86 | 0.73 | 0.06 | 0.843 | **34.14** | 2.64 |
| Wire | 2,500 | 17,469 | 6.99 | **0.74** | 0.60 | 0.795 | 3.32 | 4.76 |
| iShape | 12,870 | 175,840 | 13.66 | 0.65 | 0.42 | 0.806 | 15.84 | 6.99 |

**The similar appearance between object instances.** Instances from the same object class in iShape share similar appearance, which is challenging to embedding-based algorithms. In particular, any two object instance in iShape-Antenna, iShape-Fence and iShape-Hanger are indistinguishable according to their appearance. They are generated from either industrial standard antennas or copies of the same mesh models. Meanwhile, the appearance of objects in iShape-Branch, iShape-Log, and iShape-Wire are slightly changeable to add some variances, but appearances of different instances are still much more similar than those from other existing datasets in Table 1.

**Aspect ratio.** Table 1 presents statistics on the average aspect ratio of the object's minimum bounding rectangle for each dataset. Among them, iShape-Log's aspect ratio reaches 34.14, which is more than 10 times of other regularly shaped datasets. Such a gap is caused by two following reasons: Firstly, the shape of logs has a large aspect ratio. Secondly, partially occluding logs leads to a higher aspect ratio. iShape-Antenna also has a high aspect ratio, 9.86, which exceeds other regularly shaped datasets.

**Connected Components Per Instance (CCPI).** Larger CCPI poses a larger challenge to instance segmentation algorithms. Due to the characteristics of irregular shaped objects and the occlusion of scenes, the instance appearance under the mesh shape tends to be divided into many pieces, leading to large CCPI of iShape-Fence. As is shown in Table 1, the result on CCPI of iShape-Fence is 53.65, about 5 times higher than the second place. iShape-Branch, iShape-Hanger, and iShape-Wire also have a large CCPI that exceeds other regularly shaped datasets.

# 4   Baseline Approach

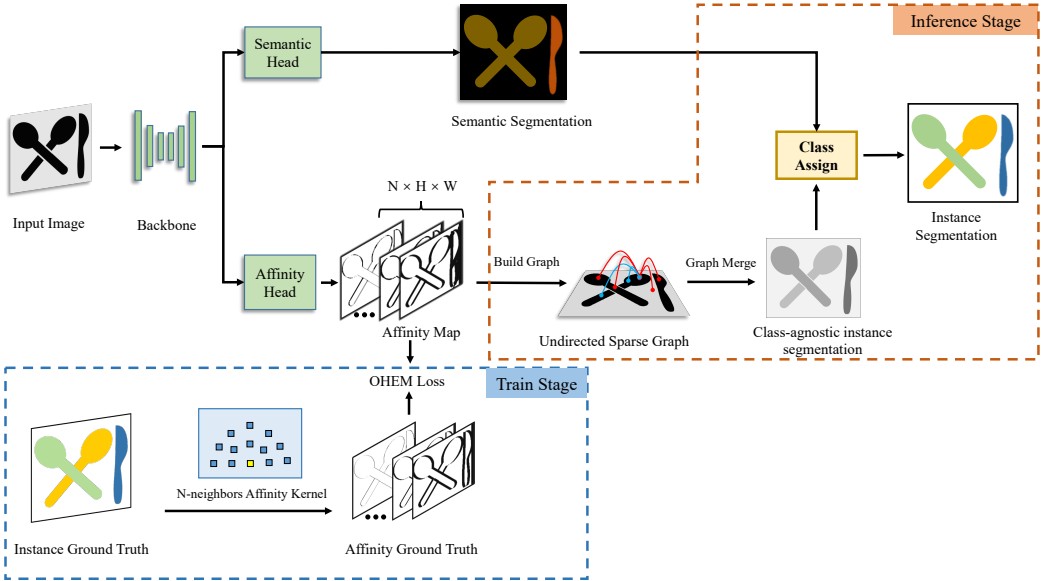

Figure 3: **Overview of ASIS**. In the training stage, the network learns to predict the semantic segmenation as well as the affinity map where the ground truth of affinity can be generated by affinity kernel and instance ground truth. In the inference stage, the predicted affinity map will be used to construct a sparse and undirected graph, with pixel as node and affinity map as edge. The final instance label then can be generated by applying a class assign module on top of the constructed graph and semantic segmentation map.

Inspired by how a person identifies a wire shown in Figure 1, We propose an affinity-based instance segmentation baseline, called ASIS, to solve Arbitrary Shape Instance Segmentation by explicitly combining perception and reasoning. Besides, ASIS includes principles of generating effective and efficient affinity kernel based on dataset property. In this section, an overview of the pipeline is firstly described in Subsection 4.1, then design principles of the ASIS affinity kernel are explained in Subsection 4.2.

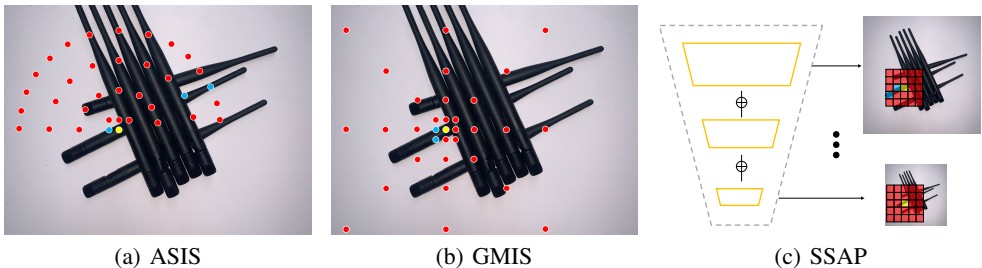

| (a) ASIS | (b) GMIS | (c) SSAP |

Figure 4: Illustration for affinity kernels. (a) ASIS affinity kernel could connect these two segments with two neighbors (blue points). (b) GMIS affinity kernel cannot reach the right segment. (c) Examples of failure case for SSAP affinity kernel. For higher resolutions (top), $5 \times 5$ affinity window cannot reach the segment on the right. For lower resolutions (bottom), the view of thin antennas are lost in the resized feature maps.

## 4.1 Overview of ASIS

As shown in Figure 3, we firstly employ the PSPNet [11] as the backbone and remove its last softmax activation function to extract features. The semantic head, which combines a single convolution layer and a softmax activation function, will input those features and output a $C \times H \times W$ semantic segmentation probability map where $C$ means the total categories number. The affinity head that consists of a single convolution layer and a sigmoid activation function will output a $N \times H \times W$ affinity map, where $N$ is the neighbor number of affinity kernel. Affinity kernel [26] defines a set of neighboring pixels that needs to generate affinity information. Examples of affinity kernels can found in Figure 4. Each channel of the affinity map represents a probability of whether the neighbor pixel and the current one belong to the same instance.

During the training stage, we apply the affinity kernel on the instance segmentation ground truth to generate the affinity ground truth. Since affinity ground truth is extremely imbalanced, an OHEM [35] loss is calculated between the predicted affinity map and the affinity ground truth to effectively alleviate the problem. For affinity map with input size $S = N \times H \times W$, we define $A = \{a_1, a_2, ..., a_S\}$ and $Y = \{y_1, y_2, ..., y_S\}$ the sets of each pixel of the predicted affinity map and the corresponding ground truth. The loss of the $i_{th}$ pixel $L_i$ is defined as:

$$L_i = -y_i \log(a_i) - (1 - y_i) \log(1 - a_i). \tag{2}$$

Assume that the set $L^{'}$ is the $Topk$ value in $L = \{L_1, L_2, ..., L_S\}$. $K$ takes the top ten percent. The OHEM loss is as follows:

$$\mathcal{L}_{aff} = \frac{1}{|L'|} \sum_{l' \in L'} l', \tag{3}$$

Affinities that connect segments of fragmented instances are important but hard to learn. Thanks to the difficulty of learning these affinities, the OHEM loss pays more attention to these important affinities. Besides, a standard cross-entropy loss for pixels $\mathcal{L}_{sem}$ is applied to semantic segmentation output. The final training loss $\mathcal{L}$ is defined as:

$$\mathcal{L} = \lambda \mathcal{L}_{aff} + (1 - \lambda)\mathcal{L}_{sem} \tag{4}$$

For the inference stage, we firstly take pixels as nodes and affinity map as edges to build an undirected sparse graph. The undirected sparse graph in Figure 3 shows an example of how a pixel node on the spoon should connect the other pixel nodes. Then, we apply the graph merge algorithm [26] on the undirected sparse graph. The algorithm will merge nodes that have a positive affinity to each other into one supernode, by contrast, keep nodes independent if their affinity is negative. Pixels that merged to the same supernodes are regarded as belonging to the same instance. In this way, we obtain a class-agnostic instance map. A class assign module [26] will take the class-agnostic instance map and the semantic segmentation result as input, then assign a class label with a confidence value to each instance.

Table 2: Qualitative results on iShape. We report the mmAP of six sub-datasets and the average of mmAP.

| Method | Backbone | Antenna | Branch | Fence | Hanger | Log | Wire | Avg |
|---|---|---|---|---|---|---|---|---|
| SOLOv2 [21] | ResNet-50 | 6.6 | **27.5** | 0.0 | 28.8 | 22.2 | 0.0 | 14.07 |
| PolarMask [15] | ResNet-50 | 0.0 | 0.0 | 0.0 | 0.0 | 18.6 | 0.0 | 3.10 |
| SE [22] | - | 38.3 | 0.0 | 0.0 | 49.8 | 20.9 | 0.0 | 18.17 |
| Mask RCNN [12] | ResNet-50 | 16.9 | 4.2 | 0.0 | 22.1 | 32.6 | 0.0 | 12.63 |
| DETR [38] | ResNet-50 | 2.1 | 2.6 | 0.0 | 32.2 | 46.2 | 0.0 | 13.85 |
| ASIS(ours) | ResNet-50 | **77.5** | 25.1 | **37.1** | **53.1** | **69.3** | **64.9** | **54.50** |

## 4.2 ASIS Affinity Kernel

Since instances could be divided into many segments, it is important to design an appropriate affinity kernel to connect those segments that belong to the same instance. As shown in Figure 4(b) and Figure 4(c), The yellow point is the current pixel. Red points belong to different instances and blue points belong to the same instance of the current pixel. The antenna that the current pixel (yellow point) belongs to has two segments that need to be connected by affinity neighbor. The previous affinity-based approaches [26, 27] don't take into account such problems and cause some failures. Hence, we propose principles of generating effective and efficient affinity kernel based on dataset property to solve Arbitrary Shape Instance Segmentation. Our affinity kernel is shown in 4(a).

Affinity kernels of GMIS and SSAP are centered symmetric, unfortunately, that will cause redundant outputs. For example, the affinity of pixel $(1, 1)$ with its right side pixel and the affinity of pixel $(1, 2)$ with its left side pixel both mean the probability of these two pixels belonging to one instance. A detailed description of redundant affinity can be found in the appendix. To reduce the network's outputs, redundant affinity neighbors are discarded in the ASIS affinity kernel. As shown in 4(a), affinity neighbors of ASIS are distributed in an asymmetric semicircle structure. Besides, the area covered by asymmetric semicircle affinity kernel is reduced by half, in other words, the demand for receptive fields is reduced, which further reduces the difficulty of CNN learning affinity.

Two main parameters determine the shape of the ASIS affinity kernel. Kernel radius $r_k$ controls the radius of the kernel and determines how far the farthest of two segments can be reached. Affinity neighbor gap $g$ represents the distance between any two nearly affinity neighbors, thus, $g$ controls the sparseness of the affinity neighbor. Since each dataset has its optimal affinity kernel, we propose another algorithm that could adaptively generate appropriate $r_k$ and $g$ based on the dataset property. Detailed descriptions of these two algorithms can be found in the appendix.

# 5 Experiments

In this section, we choose representative instance segmentation methods in various paradigms and benchmark them on iShape to reveal the drawbacks of existing methods on irregularly shaped objects. All the existing methods are trained and tested on six iShape sub-datasets with their defaults setting. And we further study the effect of our baseline method, ASIS.

**Evaluation Metrics** The evaluation metric is mainly Average Precision (AP), which is calculated by averaging the precision under mask IoU (Intersection over Union) thresholds from 0.50 to 0.95 at the step of 0.05.

**Implementation Details** The input image resolution of our framework is $512 \times 512$. The image data augmentation is flipped horizontally or vertically with a probability of 0.5. We use the ResNet-50 [36] as our backbone network and the weight is initialized with ImageNet [37] pretrained model. All experiments are trained in 4 2080Ti GPUs and batch size is set to 8. The stochastic gradient descent (SGD) solver is adopted in 50K iterations. The momentum is set to 0.9 and weight decay is set to 0.0005. The learning rate is initially set to 0.01 and decreases linearly.

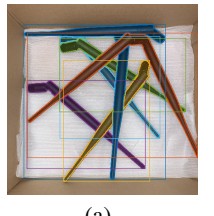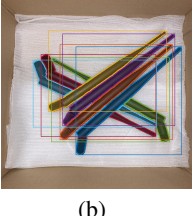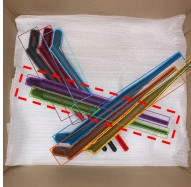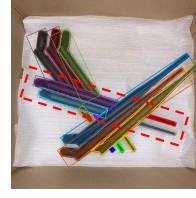

| (a) | (b) | (a) ASIS | (b) GMIS |

Figure 5: Two example false cases of ASIS on iShape-Antenna. (a) Two antennas merged into one (blue and orange). (b) ASIS fails to connect the right parts of an object (red and sky blue).

Figure 6: Results compared with GMIS kernel. As shown in (b), GMIS fail to connect segments that belong to one instance.

## 5.1 Experiment Results

We evaluate the proposed ASIS and other popular approaches on iShape. The quantitative results are shown in Table 2 and some qualitative results are reported in Figure 7.

As is shown in Table 2, the performance of Mask R-CNN [12] is far from satisfactory on iShape. We think the drop in performance mainly comes from three drawbacks of the design. Firstly, the feature maps suffer from ambiguity when the IoU is large, which is a common characteristic of crowded scenes of irregular shape objects. Also, Mask R-CNN depends on the proposals of RoI, which may be abandoned by the NMS algorithm due to large IoU and lead to missing of some target objects. Moreover, many thin objects can not be segmented by Mask R-CNN because of its RoI pooling, which resizes the feature maps and lost the view of thin objects. The recent proposed end-to-end object detection approach, DETR [38], shake of the reliance of NMS and can better deal with objects with large IoU and achieve better performance, as shown in the table. However, DETR still suffers from the RoI pooling problems and performs badly on thin objects, as shown in Figure 7.

We also report some qualitative results of SE [22] in Figure 7. As is shown in the figure, one common failure case of SE is that when the length of irregular objects is longer than a threshold, the object will be split into multi instances, for example, the wire in Figure 7. We think that's because SE will regress a circle of the target instance and then calculate its IoU with the mask for supervision. However, for long and thin irregular objects, the radius of the center circle can not reach the length of the target object, leading to a multi-split of a long instance. Also, instances that share the same center may cause ambiguity to SE, such as hanger and fence in Figure 7. Moreover, many centers of irregular objects lie outside the mask, making it hard to match them to the objects themselves.

We evaluate SOLO v2 [21] on the proposed iShape and find that it failed to segment instances that share the same center, for example, fences in Figure 7. Also, since SOLO V2 depends on the center point as SE, it also suffers from performance drop caused by object centers that lie outside the mask.

In Table 2, we report the performance of PolarMask [15] on our dataset. As is shown in the table, PolarMask can not solve the instance segmentation of irregular objects. That is because PolarMask can only represent a thirty-six-side mask due to its limited number of rays. Hence, it can not handle objects with hollow, for example, the fences. Also, they distinguish different instances according to center regression, which, however, can not handle instances that share the same center. We also find that PolarMask can only tackle some cases of logs in iShape, which looks like circles on the side and fit its convex hull mask setting.

Thanks to the perception and reasoning mechanism as well as the well-designed affinity kernels of our ASIS, it obtained the best performance on iShape. In Table 2, ASIS advances other approaches by 36% on the mmAP metric. However, there are still some drawbacks to the design of ASIS and some failure cases caused by them. For example, in Figure 5(a), two instances are merged into one. We think that's because the graph merge algorithm is a kind of greedy algorithm, while the greedy algorithm makes optimal decisions locally instead of looking for a global optimum. Hence, ASIS is not robust to false-positive (FP) with high confidence. Also, ASIS fails to connect the two parts of an object if they are far away from each other, for example, the antenna on Figure 5(b). We think that's because CNN is not good at learning long-range affinity.

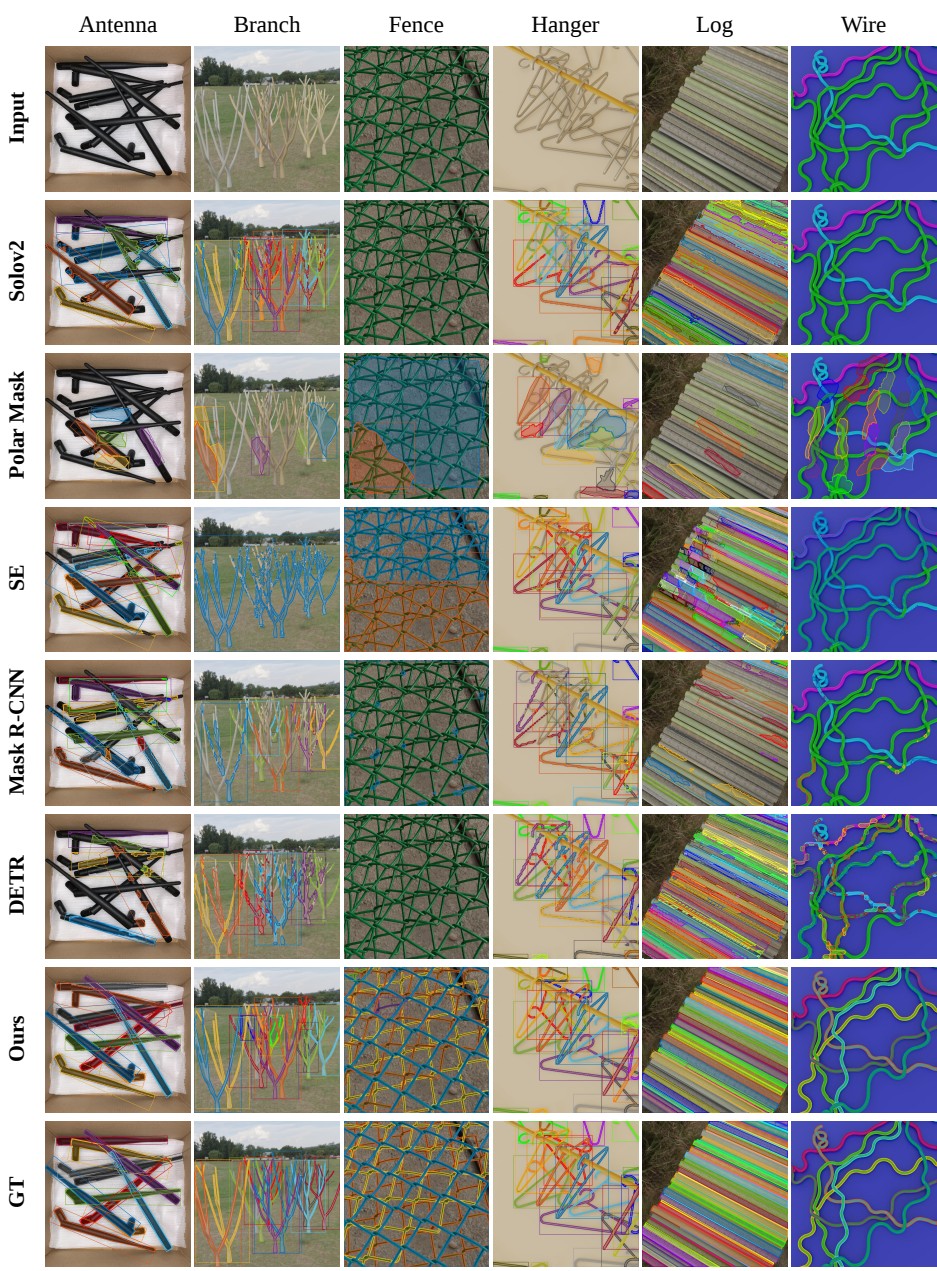

Figure 7: Qualitative results of different instance segmentation approaches on iShape.

## 5.2 Ablation Study

**Effect of ASIS.** We study the effect of ASIS in Table 3, where ASIS advances GMIS by 10.9% on iShape-Antenna. We think that is because our well designed affinity kernels based on dataset property can better discover the connectivity of different parts of an object. While GMIS suffers from its sparsity in distance and angle, results are shown in Figure 6(b). We also use ground truth affinity map to explore the upper bound of ASIS, where a 98.5% mAP is achieved, showing its great potential. Moreover, we find our non-centrosymmetric design of affinity kernels outperform centrosymmetric ones in the table. We think such a design cut off the output and calculation redundancy and reduce the requirement of large receptive field from CNN, simplifying representation learning.

**Effect of OHEM.** Table 3 shows that OHEM boosts the performance of GMIS and ASIS by a large margin. We think that is because OHEM can ease problems caused by imbalance distribution of positive and negative affinity.

Table 3: Comparison result of GMIS and ASIS. "SY" and "ASY" indicate a centrosymmetric or asymmetric affinity kernel respectively. $\sqrt{}$ denotes equipped with and ∘ not.

| Affinity Kernel | Neighbors | Affinity GT | OHEM | mAP |
|---|---|---|---|---|
| GMIS [26] | 56 (SY) | ∘ | ∘ | 44.5 |
| | | ∘ | $\sqrt{}$ | 69.9 |
| | | $\sqrt{}$ | - | 90.2 |
| | 28 (ASY) | ∘ | $\sqrt{}$ | 72.7 |
| ASIS(ours) | 53 (ASY) | ∘ | ∘ | 58.4 |
| | | ∘ | $\sqrt{}$ | 77.5 |
| | | $\sqrt{}$ | - | 98.5 |

# 6   Conclusion

In this work, we introduce a new irregular shape instance segmentation dataset (iShape). iShape has many characteristics that challenge existing instance segmentation methods, such as large overlaps, extreme aspect ratios, and similar appearance between objects. We evaluate popular algorithms on iShape to establish the benchmark and analyze their drawbacks to reveal possible improving directions. Meanwhile, we propose a stronger baseline, ASIS, to better solve Arbitrary Shape Instance Segmentation. Thanks to the combination of perception and reasoning as well as the well-designed affinity kernels, ASIS outperforms the state-of-the-art methods on iShape. We believe that iShape and ASIS can serve as a complement to existing datasets and methods to promote the study of instance segmentation for irregular shape as well as arbitrary shape objects.

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
