# OpenReview forum: "iShape: A First Step Towards Irregular Shape Instance Segmentation"
_NeurIPS.cc/2021/Track/Datasets_and_Benchmarks/Round1 — Submitted to NeurIPS 2021 Datasets and Benchmarks Track (Round 1)_

### Official Review · Reviewer_12ZL · 2021-07-04
**Review of Irregular Shape Instance Segmentation**

**Rating:** 5
**Confidence:** 3
**Correctness:** Yes.
**Clarity:** Yes, the paper is with good formats a…

**Strengths:**

+ Good paper writing and formatting.

+ The proposed dataset is interesting. It focuses on the real scenarios in industrial applications where prior state-of-the-art methods are shown to fall short.

+ The rationale behind the method design is clearly stated and the proposed components have been empirically shown effective with adequate analysis and explanations.


**Weaknesses:**

1. The proposed framework leverages the affinity groudtruth for training guidance while the other methods do not. The author can try to apply the affinity loss with OHEM strategy to other state-of-the-art methods to show the effectiveness.

2. The model seems to be a general framework for instance segmentation. The author could apply the method to other popular datasets for instance segmentation to have a comprehensive comparison and show its generalization ability.


**Additional Feedback:**

[Post Rebuttal] The authors' rebuttal does not address some of my concerns.
OHEM is a popular trick that could bring considerable improvements, which has been widely verified in the vision community, and OHEM is not one of the contributions in this paper. Only applying OHEM on the proposed method causes an unfair comparison with the others. Also, it is not proper to use the default configurations to training other methods in your setting because even the same algorithm performs very differently on different benchmarks. Directly applying the default setting to another dataset usually leads to inferior performance, according to the reviewer's experience. The unfair comparison does not tell the superiority of the method and the necessity of the benchmark.

Considering other issues raised by Reviewer#2, the reviewer believes that this dataset is somewhat unrealistic since there are several inherent constraints listed by Reviewer#2. Also, as claimed by the authors, the dataset is a complement to the existing dataset. It would be better to verify the proposed method on other popular benchmarks.

**Documentation:**

Yes, the dataset has clear and detailed documentation.

**Ethics:**

I think no.

**Relation To Prior Work:**

Yes.

**Summary And Contributions:**

This paper proposes a new benchmark for tackling irregular objects in instance segmentation.

Prior arts fail to achieve promising performance on the proposed benchmark, and the authors propose a new method called ASIS that leverages the semantic segmentation and affinity predictions for yielding the final accurate instance segmentation results.

---

> ### Author Response · Authors · 2021-07-13
> **Reply to Reviewer4**
>
> Thanks for your appreciation of our work. All comments are summarized and addressed as follows.
>
> **Q1: The author can try to apply the affinity loss with OHEM strategy to other state-of-the-art methods.**
> Thank you for the suggestion, we didn't add affinity and OHEM loss for other methods for two reasons:
> - Due to different paradigms of other methods, applying affinity and OHEM loss on other methods is not obvious, even difficult.
> - Our purpose is to benchmark other methods on iShape. Thus, it is better to use the default setting of other methods, and not modify their model or setting.
>
> **Q2: The author could apply the method to other popular datasets for instance segmentation.**
> According to Table.1, popular datasets like COCO and Cityscapes have almost no irregular characteristics. Due to our ASIS is based on GMIS and designed especially for irregular shape, the mmAPs of ASIS on such datasets are almost similar to GMIS (https://arxiv.org/abs/1811.10870)

---

### Official Review · Reviewer_Qvwj · 2021-07-04

**Rating:** 6
**Confidence:** 3

**Strengths:**

The paper proposed a novel dataset for evaluating instance segmentation algorithms, and focuses on six challenging aspects of irregular objects. The dataset is released, well-documented, and easy to access.
An evaluation of several existing algorithms on the new dataset is presented, and a novel instance segmentation method is proposed.


**Weaknesses:**

I found the paper easy to read and understand, however, it could be organized better. For example, the details of the data creation are currently in supplementary material, and some of them need to be in the main paper. It appears that part of the data is generated synthetically (using Blender), which part is collected from real scenes. This process, as well as main details of data generation need to be added to the main manuscript. Also, given that there is both synthetic and real data, how does the source of data affect performance of existing techniques? How realistic is the synthetic data?

Also, the authors propose a novel instance segmentation algorithm. How is is similar and different from existing works? Based on what properties are the other baseline methods selected for evaluation? In Table 2, the proposed method outperforms on all data subsets, except for branch. Why does this happen?


**Additional Feedback:**

None

**Clarity:**

In general, the paper is well, written. However, important details are missing. Please see list below:

- What does ASIS, the name of the proposed method, stand for?
- Why is aspect ratio an important object property that needs to be discussed? Is it not sensitive to image rotation, whereas modern algorithms often use random rotations as a form of data augmentation.
- Table 1: for the proposed dataset, please also include summary statistics (e.g., average), to be more comparable to other datasets. Please also add reference to other datasets’ papers
- Figure 3 is hard to read, please increase font size
- Table 2: why is evaluation performed only using ResNet-50 backbone? Will results change with a different backbone?
- Figure 6: can you please add zoom ins, or otherwise highlight the main differences of the methods in the figure? The differences are described in text, but difficult to extract from the figure.
- In the data release, when I try to access instance maps in shape/ishape_dataset/antenna/train/instance_map/***.png, I see black images only.  Could you please provide some source code to explain how to load the instance maps and/or add visualizations of the instance-segmented images into the data repository?

Typos:
- Line 151: “segemtation” -> segmentation
- Line 153: “As is shown in the table” -> add link to table
- Line 170: “stage, We apply” -> stage, we apply
- Line 233: “lost the view of thin objects” -> bad grammar
- Line 245: “SOLO V2” -> which method is that? Please add a reference.
- Line 248: “PolarMask” -> which method is that? Please add a reference.
- Line 249: “That’s because” -> That is because

**Correctness:**

It appears that the data is constructed synthetically using blender and collected from other datasets. While the details of the synthetic data are outlined in supplementary material, the details of manually annotated data subsets ( supplementary material, 3.1) are missing. Who performed these annotations, and how was this process organized?


**Documentation:**

- +Data is hosted on Github (can be downloaded) and will be released on Kaggle later. Dataset seems easy to access and use
- -Some important details of data collection (details of manual annotation, maintenance, and ethical and responsible use) are missing


**Ethics:**

No concerns.

**Relation To Prior Work:**

Related work is well-written, but please explain what are the current state-of-the-art methods.


**Summary And Contributions:**

The paper presents iShape, a benchmark of 2D instance segmentation for irregular shaped objects (e.g., antenna, branch, fence) and benchmarks existing algorithms on the new data. The authors also propose ASIS, their own instance segmentation algorithm. Segmenting instances of irregular shapes is an important problem in many applications of machine learning and computer vision, such as medical imaging, astronomy, etc. Thus, releasing the benchmark would spur additional interest in development of novel methods for these fields.

---

> ### Author Response · Authors · 2021-07-13
> **Reply to Reviewer3**
>
> Thanks for your meticulous comments. Below you will find our responses to your comments.
>
> **Q1: Data creation should in the main paper.**
> We have added data creation to the main paper in the new version.
>
> **Q2: How does the source of data affect performance of existing techniques?**
> Our dataset focuses on the evaluation of instance segmentation methods on irregularly shaped objects, for which the synthetic images can serve well. In our setting, models trained on synthetic data will test on synthetic data too. Thus data source has almost no effect to result.
>
> **Q3: How realistic is the synthetic data?**
> Although realistic is not the most important factor for evaluating models, we still try to make our synthetic data more realistic by applying 400+ realistic background and lighting environments, physic engine, and using a ray-tracing render engine to render the RGB image. The result image example can be found online: http://47.103.201.240:9000/ishape/ishape_dataset/log/train/vis/
>
> **Q4: How is ASIS similar and different from existing works?**
> The ASIS is based on the previous method GMIS and designed especially for irregular shapes. Difference between ASIS and GMIS includes:
> - ASIS has a well-designed semicircle structure affinity kernel. GMIS has a square structure kernel.
> - We employ an OHEM loss to ease problems caused by the imbalance distribution of positive and negative affinity.
>
> We study the effect of those differences in Table 3
>
> **Q5: What properties are the other baseline methods selected for evaluation?**
> We chose the representative instance segmentation method in various paradigms, e.g. one-stage, two-stage and bottom-up paradigm.
>
> **Q6: Why does ASIS not outperforms on branch?**
> ASIS results on branch include many parts that have not been merged. Those parts lead to a large number of small FPs, which causes serious performance drop (e.g. https://ishape.github.io/image/vis_tmp/asis/branch/107.jpg ).
>
> **Q7: Details of manually annotated data subsets are missing**
> We will add more details in dataset creation.
>
> **Q8: What does ASIS, the name of the proposed method, stand for?**
> From our abstract:
> > ASIS explicitly combines perception and reasoning to solve **A**rbitrary **S**hape **I**nstance **S**egmentation including irregular objects.
>
>
> **Q9: Why is aspect ratio an important object property that needs to be discussed?**
> In our paper, the aspect ratio is the object’s minimum bounding **rotatable** rectangle which is not axis aligned. And it is not sensitive to image rotation. If the aspect ratio of one object is large, the object should be irregularly shaped. Thus, the aspect ratio is an important object property that needs to be discussed.
>
> **Q10: Table 2: why is evaluation performed only using ResNet-50 backbone? Will results change with a different backbone?**
> Thank you for the suggestion, we will add more experiments with different backbones in the next version.
>
>
> **Q11: Add visualizations of the instance-segmented images**
> We offer visualizations of the instance-segmented images online. http://47.103.201.240:9000/ishape/ishape_dataset/antenna/val/vis/
>
> **Other typos.**
> Other typos have been fixed  in the new version.

---

> > ### Comment · Reviewer_Qvwj · 2021-07-14
> > **Response**
> >
> > Thank you for your explanations! The links to the data and explanations clarify my original questions.

---

### Official Review · Reviewer_SpE4 · 2021-07-05
**Interesting dataset, not well motivated**

**Rating:** 5
**Confidence:** 4
**Clarity:** The paper is overall well written and…

**Strengths:**

The paper is well written and easy to follow. The key statistics of the dataset are well explained. The addition of a strong baseline model provides a good starting point for future research.

The reasoning of why common segmentation methods such as Mask RCNN fail in the proposed tasks is convincing.


**Weaknesses:**

Major:

1. A formal definition of “irregular shape” is not explicitly given. While the paper describes a range of statistics for several popular datasets and points out the differences, it does not provide a clear definition of shape irregularity. One could argue that the logs in the logs dataset are regularly shaped given their stereotypical color and rectangular shape. In contrast, many classes in the iSAID such as harbors have irregular shapes.

2. The motivation of an image dataset with only irregularly shaped objects is not well explained. It is unclear why having a dataset with only a few classes of seemingly arbitrarily chosen classes of irregularly shaped objects is helpful. The Antenna class and Log class seem to have similar long rod-like shapes, it seems redundant to have these two. Also the objects seem to have more or less homogeneous colors. One could provide concrete examples of potential real-world use cases to justify the usefulness of such a task.

3. In many datasets such as COCO and iSAID, multiple classes are often present in the same image. In contrast, the proposed dataset limits the number of classes in an image to one, which is unrealistic in most real-world vision tasks. The authors fail to justify the decision of not allowing more than one class in an image.

4. 5 of 6 sub datasets are synthetic, however this is not highlighted as there is only one mention of “synthetic” in the entire paper. Even among synthetic images, this is visually simplistic compared with some of the more sophisticated synthetic datasets such as GTA dataset (https://arxiv.org/pdf/1608.02192.pdf). It is well known that there is usually a generalization gap between synthetic and real images. As a result, it is unclear how much of what we learned with this dataset will generalize to real images.

Minor:

5. Standard deviation is missing from results.

6. GitHub missing instructions and code to reproduce results.


**Additional Feedback:**

Since the authors claim that the proposed dataset is “a complement to existing datasets”, it seems reasonable to clarify whether the proposed ASIS method only works for the proposed dataset or it also generalizes to common segmentation datasets like COCO.

**Correctness:**

As discussed above, several choices of the construction of the dataset are questionable.

The evaluation methods are mostly appropriate.


**Documentation:**

The documentation on Github is insufficient. No instructions/example commands to reproduce the main results in the paper have been made available at https://ishape.github.io/ .

**Relation To Prior Work:**

The paper has detailed discussions of prior work.

**Summary And Contributions:**

The paper proposes a mostly synthetic image dataset for instance segmentation of irregularly-shaped objects. The authors also propose a strong baseline method for irregular shape segmentation and reason why popular segmentation methods fail.

---

> ### Author Response · Authors · 2021-07-13
> **Reply to Reviewer2**
>
> Thanks for your valuable comments. All comments are summarized and addressed as follows.
>
> **Q1: A formal definition of “irregular shape”.**
> Besides the statistics in Table 1, we also describe irregular shape in the introduction section:
> > However, most of them focus on regularly shaped objects and only a few study irregular ones, which are thin, curved, or having complex boundaries and can not be well-represented by regularly rectangular boxes.
>
> A more formal definition of irregular shape in instance segmentation is "Area of the bounding box is far large than the area of instance mask or aspect ratio of the bounding box is large (e.g. >5)".
>
> **Q2: One could argue that the logs in the logs dataset are regularly shaped.**
> The average aspect ratio of the bounding box in iShape-log is 34.14, which belongs to irregular shape according to our definition that "aspect ratio of the bounding box is large".
>
> **Q3: Many classes in the iSAID such as harbors have irregular shapes.**
> We agree that many classes in the iSAID have irregular shapes, but we argue that:
> - There is almost no overlap between instances in iSAID, which is a common case in remote sensing images and cannot represent most instance segmentation scenes.
> - Only a few instances are of irregular shape, and most of the irregular shapes are long rod-like shapes, and the types of irregular shapes are not rich enough.
>
> **Q4: The motivation of an image dataset with only irregularly shaped objects is not well explained.**
> As Reviewer 3 said, Segmenting instances of irregular shapes is an important problem in many applications. Our dataset focuses on the evaluation of instance segmentation methods on irregularly shaped objects. Thus, iShape only considers irregularly shaped objects.
> Examples of real-world applications:
> - The antenna is the actual demand we encountered. We need to count the antenna products on the production line and grab them by robot arm which needs instance segmentation for irregular shape.
> - Robots move wood in the logging yard or steel pipes in the workshop.
>
>
> **Q5: The Antenna class and Log class seem to have similar long rod-like shapes, it seems redundant to have these two.**
> Antenna and Log belong to different irregular shape scenarios. The Log scene is derived from robots moving wood in a logging yard or steel pipes in the workshop. In Log scene, the camera is not fixed, and objects are neatly stacked in the same direction. In Antenna scene, the camera is fixed, and objects have more freedom in direction.
>
> **Q6: Limits the number of classes in an image to one.**
> - Lots of real-world vision tasks only need one class, e.g. pedestrian detection, face detection, cell instance segmentation, and industrial standard product detection.
> - Each method performs differently in different irregular scenarios. Different sub-datasets of iShape represent different irregular shape scenarios that can help algorithms to analyze their weaknesses on shape generalization.
> - We pay more attention to the ability of the algorithm to distinguish irregular shape objects, rather than the classification ability.
>
> **Q7: Generalization gap between synthetic and real images.**
> Our dataset focuses on the evaluation of instance segmentation methods on irregularly shaped objects, for which the synthetic images can serve well. Our synthetic setting is based on works like "Playing for Benchmarks" and "Virtual Worlds as Proxy for Multi-Object Tracking Analysis", which indicate that synthetic data can be used as an effective way to evaluate models.
>
> **Q8: Standard deviation is missing from results.**
> We will add Standard deviation for all methods in the next version.
>
> **Q9: GitHub missing instructions and code to reproduce results.**
> We have released config and models in the experiments on https://ishape.github.io/#5-benchmark
>
>
> **Q10: clarify whether the proposed ASIS method only works for the proposed dataset or it also generalizes to common segmentation datasets like COCO.**
> According to Table.1, popular datasets like COCO and Cityscapes have almost no irregular characteristics. Due to our ASIS is based on GMIS and designed especially for irregular shape, the mmAPs of ASIS on such datasets are almost similar to GMIS(https://arxiv.org/abs/1811.10870)

---

### Official Review · Reviewer_o7ny · 2021-07-05
**Nice paper and dataset but with some concerns**

**Rating:** 6
**Confidence:** 3

**Strengths:**

Strength of this paper:
1. The proposed dataset looks good, and can be used as an additional benchmark for all future instance segmentation methods.
2. The authors tested multiple benchmarks on their dataset, and showed that existing methods do not work well. This verifies my concern on previous papers too.
3. The proposed ASIS framework uses a more effective affinity kernel approach and it improves the performance quite drastically.

**Weaknesses:**

Concerns that I hope authors can help me here:

(1) For the dataset:

1. Most of the data (or all of them) are string- or bar- like objects. I know the authors trying to convey the message that this is irregular and challenging, but I'm not sure if this covers the definition of 'irregular shapes'. For example, you can have irregular surfaces like leafs with different shapes and appearances, and so on. I would suggest tuning the topic and title to a more specific one, if the dataset cannot cover all major irregular cases.
2. It looks to me most of the data are synthetic, maybe rendered by graphics. Do you think this may be a problem when transferring the trained model like ASIS on iShape to real world data?

(2) For the ASIS:

3. The affinity kernel comparison in Figure 4 is confusing me. Based on my understanding the affinity modeling capability is affected by both the kernel shape and resolution. And 4(a) ASIS seems to be more fine-grained than 4(b) GMIS. Is this fair? Also, the ASIS kernel is semicircle since authors believe the other half is redundant. But do you really guarantee that the missed connections can be recovered when the kernel is moving to another location especially when you only have a sparse kernel? And the kernel parameters r_k and g must be pre-defined before training (not learnable)?
4. Can the sparse graph contain loops or not? Details are quite missing on the graph part.
4. Paper structure: the ASIS section spends quite many texts on the loss. But I don't think that's the correct focus. I would suggest adding more description on how you build the sparse graph, and how you do the graph merge. In addition, I suggest adding two or three sentences (or one formula) introducing the affinity kernels for general readers.

(3) For experiments:

5. Maybe I missed this detail, but did you re-train all the previous methods on your iShape data? This needs to be explicitly stated otherwise it's unfair. Also, could you compare the cost such as memory (or #params) and time? This can help readers rule out the gain from the increased network capacity.
6. Since the authors were focusing on affinity kernels, I would like to see some kernel or graph visualizations in the results compared to other affinity kernels. But unfortunately there is not. This is critical because it verifies the design.

**Additional Feedback:**

Please check the weakness comments for details.

[Post rebuttal comments]
I'm ok to accept this now, but please fix the comments I added in my response.

**Clarity:**

The paper is well written in most parts. The ASIS (section 4) needs some adjustment, like describing more on graph construction and merging, and maybe give a short introduction on affinity kernels.

**Correctness:**

Most of the claims in the paper are correct and sound. Some claims such as 'irregular shapes' may need further consideration as mentioned in the weakness. And in the experimental section, authors need to explicitly state how they re-train the baselines and make sure the comparison was completed on similar total network capacity.

**Documentation:**

Yes I think so. I'm not sure if there is any documentation or user instruction for their dataset, but I would highly suggest to do so rather than only releasing the raw data alone. For benchmarks I'm not sure but I hope authors can release them together with their models.

**Ethics:**

No I don't think there is any problem since the data is mostly synthetic.

**Relation To Prior Work:**

Yes I believe so. Studying shape segmentation on irregular shapes like this is well explained and differs from previous works clearly.

**Summary And Contributions:**

In general, this paper addresses an important problem in shape segmentation: the irregular cases. Previously I have tried to push many papers to provide results on irregular input shapes, but most of them never took my suggestions seriously, and I get bored of seeing chairs, tables, and airplanes. So I'm pretty happy that finally this problem is being researched.

Overall I like this submission, but I do have concerns as listed below. So I decide to weak accept.

---

> ### Author Response · Authors · 2021-07-13
> **Reply to Reviewer1**
>
> Thanks for your thoughtful comments. Below you will find our responses to your comments.
>
> **Q1: I would suggest tuning the topic and title to a more specific one, if the dataset cannot cover all major irregular cases.**
> Although we tired our best to provide six different typical irregular shape scenes, it is indeed hard to say that iShape covers all major irregular cases. Hence, we rephase the title to "iShape: A First Step Towards Irregular Shape Instance Segmentation".
>
> **Q2: Maybe a problem when transferring the trained model on iShape to real-world data?**
> Our dataset focuses on the evaluation of instance segmentation methods on irregularly shaped objects, for which the synthetic images can serve well. Our synthetic setting is similar as pervious works like "Playing for Benchmarks" and "Virtual Worlds as Proxy for Multi-Object Tracking Analysis", which indicate that synthetic data can be used as an effective way to evaluate models.
>
> **Q3: Figure 4 is confusing.**
> Sorry for the confusion that we set a wrong resolution of that image by a mistake when adjusting the diagram. We have fixed this problem and the resolutions of ASIS and GMIS are the same now.
>
> **Q4: Guarantee that the missed connections can be recovered in ASIS.**
> For the concern of sparse kernel, although the affinity kernel of ASIS is more effective and efficient than GMIS, it still cannot guarantee that the missed connections can be recovered. According to Table.3, mAP is 98.5% when using affinity GT. For the concern of the semicircle design, we give detailed illustrations in Section 1.3 and Figure 1 in our appendix. We've also checked that the mAP of the full circle and semicircle affinity kernel achieves the same mAP, 98.5% with affinity GT.
>
> **Q5: Kernel parameters $r_k$ and $g$ must be pre-defined before training?**
> For our experiments, ASIS has pre-defined kernel parameters which were calculated from the training set. A learnable affinity kernel is interesting direction which is worth to explore as future work.
>
> **Q6: Can the sparse graph contain loops or not?**
> The sparse graph contains loops. For example, the Figure 1(b) in the appendix, when kernel at point $C$, $AC$ and $AD$ are connected. When kernel moves to point $D$, $CD$ is connected. Thus, there is a loop $ACD$ in the sparse graph.
>
> **Q7: Details are quite missing on the graph part.**
> Thanks for pointing out, we have added more descriptions on graph construction and merging. We also have added a short introduction on affinity kernels.
>
> **Q8: Did you re-train all the previous methods on your iShape data?**
> Yes, all the previous methods were trained and tested on six iShape sub-datasets with their defaults setting. We have added explicitly statement about the re-training in the experiments section.
>
> **Q9: Documentation or user instruction for their dataset**
> We have added documentation to the dataset.
>
> **Q10: For benchmarks I'm not sure but I hope authors can release them together with their models.**
> We have released configs and models in the experiments on https://ishape.github.io/#5-benchmark

---

> > ### Comment · Reviewer_o7ny · 2021-07-20
> > **Response**
> >
> > Thanks for the clarification.
> > I think the paper is ok to be accepted now, although authors seem to misunderstand some of my questions:
> > 1. What I mean kernel-visualization is to draw the kernel weights (like CNN) so we can understand if the learned kernels are following specific patterns or learning something meaningful, not simply comparing the results as in Figure 6.
> > 2. The memory and inference time information seem to be still missing.
> >
> > Please try to fix this for the final version.

---

> ### Author Response · Authors · 2021-07-14
> **Reply to Reviewer1 - Part 2**
>
> **Q11: Some kernel or graph visualizations in the results.**
> We have added Figure 6 in the new version, the visualizations of both ASIS and GMIS results.

---

### Author Response · Authors · 2021-07-13
**Change log of the revised version**

We have updated our paper based on the reviewer's comments, note that updates are highlighted by yellow color in the paper pdf file:
1. Rephase the title to "iShape: A First Step Towards Irregular Shape Instance Segmentation".
1. Move "data creation" to the main paper, and add details of manually annotated data. (R3)
1. We have added more descriptions on graph construction and merging. We also have added a short introduction on affinity kernels. (R1)
1. Fix the wrong resolution in Figure 4. (R1)
1. Add summary statistics of iShape in Table 1 (R3)
1. Add more details about experiments (R1, R3)
1. Fix typos listed by R3

---

> ### Author Response · Authors · 2021-07-14
> **Additional update**
>
> 1. Add Figure 6 for visualizations in the results compared to other affinity kernels (R1)
> 1. Increase font size of Figure 3 (R3)

---

### Decision · Program_Chairs · 2021-07-27

**Decision:**

Reject

**Comment:**

This paper received conflicting reviews. AC and PC discussed the paper. We agree that the proposed dataset is filling in an interesting gap in instance segmentation study. However, the model study is not comprehensive enough yet to convince all the reviewers the problem can not be solved by the existing methods. We suggest the authors spend some more time polishing the experiments and paper for later submissions.